# Tucum Fiber from Amazon *Astrocaryum vulgare* Palm Tree: Novel Reinforcement for Polymer Composites

**DOI:** 10.3390/polym12102259

**Published:** 2020-10-01

**Authors:** Michelle Souza Oliveira, Fernanda Santos da Luz, Andressa Teixeira Souza, Luana Cristyne da Cruz Demosthenes, Artur Camposo Pereira, Fabio da Costa Garcia Filho, Fábio de Oliveira Braga, André Ben-Hur da Silva Figueiredo, Sergio Neves Monteiro

**Affiliations:** 1Military Institute of Engineering, IME, Praça General Tibúrcio 80, Urca, Rio de Janeiro 22290-270, Brazil; fsl.santos@gmail.com (F.S.d.L.); andressa.t.souza@gmail.com (A.T.S.); eng.luanademosthenes@gmail.com (L.C.d.C.D.); camposo.artur@gmail.com (A.C.P.); abenhur@ime.eb.br (A.B.-H.d.S.F.); snevesmonteiro@gmail.com (S.N.M.); 2Department of Mechanical and Aerospace Engineering, University of California San Diego—UCSD, La Jolla, CA 92093, USA; fdacostagarciafilho@eng.ucsd.edu; 3Department of Civil Engineering, Federal Fluminense University—UFF, Niterói 24210240, Brazil; fabiobraga@id.uff.br

**Keywords:** tucum fibers, polymer composites, Izod, ballistic armor, pullout, natural fiber composites

## Abstract

The replacement of synthetic fibers by natural fibers has, in recent decades, been the subject of intense research, particularly as reinforcement of composites. In this work, the lesser known tucum fiber, extracted from the leaves of the Amazon *Astrocaryum vulgare* palm tree, is investigated as a possible novel reinforcement of epoxy composites. The tucum fiber was characterized by pullout test for interfacial adhesion with epoxy matrix. The fiber presented a critical length of 6.30 mm, with interfacial shear strength of 2.73 MPa. Composites prepared with different volume fractions of 20 and 40% tucum fiber were characterized by tensile and Izod impact tests, as well as by ballistic impact energy absorption using .22 ammunition. A cost analysis compared the tucum fiber epoxy composites with other natural and synthetic fiber reinforced epoxy composites. The results showed that 40 vol% tucum fiber epoxy composites increased the tensile strength by 104% and the absorbed Izod impact energy by 157% in comparison to the plain epoxy, while the ballistic performance of the 20 vol% tucum fiber composites increased 150%. These results confirmed for the first time a reinforcement effect of the tucum fiber to polymer composites. Moreover, these composites exhibit superior cost effectiveness, taking into account a comparison made with others epoxy polymer composites.

## 1. Introduction

Since the beginning of the 21st century, official environmental regulations have motivated researchers all over the world to pursue solutions regarding energy saving, climate changes and pollution control. In this respect, the use of renewable materials and the consolidation of the circular economy play a major role toward the development of a sustainable society. Today, an important example is the use of natural lignocellulosic fibers (NLFs) to replace synthetic fibers, which has become the object of intense research, especially when used as reinforcement for polymer matrix composites [1,2,3,4,5,6,7,8,9,10,11]. These NLFs, composed mainly of lignin and cellulose, have innumerous advantageous characteristics, such as low density and cost effectiveness, as well as biodegradability and worldwide abundance [12,13]. Moreover, several different NLFs have been reported to be successfully applied as composite reinforcement for a wide range of engineering applications [14,15] including building construction [16,17], the automotive industry [18] and ballistic armor [19,20,21,22]. Additionally, from the processing aspect, NLFs cause less damage to equipment compared to synthetic fibers, such as the glass fiber [12,23].

Among the NLFs, a lesser known fiber, the tucum fiber in Figure 1, extracted from the leaf-stalk (petiole) of the *Astrocaryum vulgare* palm tree, endemic of the Amazon region in South America, is the object of the present work. As schematically illustrated in Figure 1a, the palm genus *Astrocaryum* with 40 species is common in tropical South America extending northwards up to Central America. In Figure 1a, the digit inside the square is the number of *Astrocaryum* species in corresponding countries, which reveals the predominance of this plant in distinct South American regions. Twenty-six species are native of Brazil (26), as well as present in Peru (14), Colombia (11), Guyana (9), Suriname (9), Bolivia (8), French Guiana (8), Venezuela (6), Ecuador (4), Costa Rica (2), Panama (2), and Trinidad (1). In Brazil, the tucum palm tree is also known as chambira palm, in Colombia and Equador as corombolo or palm-coconut [24,25,26,27].

The species *Astrocaryum vulgare* of tucum palm tree, Figure 1b, is one of the most important for the indigenous communities of the northwestern Amazon. In recent decades, tucum-based products have gained acceptance among tourist and craft stores, becoming an important cash crop for local families. The fibers obtained from its unexpanded leaves and stalks are used to make a great variety of products for daily needs such as fishing nets, and rope for tying the canoe, as well as items for ornaments, utensils, hats, bags and hammocks. Although tucum fiber has long been popular in the Colombian Amazon, only in the last few years it has been introduced into the most important craft fairs in Bogota [28,29]. The leaf-stalks of the tucum palm, Figure 1c, have nowadays become a potential source of vegetal fiber. However, to the knowledge of the authors of the present work, there has never been any publication about composites reinforced with tucum fiber.

Therefore, for the first time, this study aimed to perform an investigation on basic characteristics of the tucum fiber in terms of its density, pullout properties, microstructural aspects, and their use as polymer matrix composites subjected to tensile tests as well as under impact conditions related to Izod and ballistic tests. It is important to mention that the present study brings unprecedented data, in ballistic performance. In particular, it compared the obtained parameters with others from commonly studied NFLs composites in ballistic application, as well as related cost. The tucum fiber morphology is analyzed by scanning electron microscopy (SEM).

## 2. Materials and Methods

### 2.1. Materials

The *Astrocaryum vulgare* tucum fibers were obtained from a local market in the state of Amazonas, Brazil. These fibers were dry and gained a visual whitish appearance. Figure 1c illustrates a bunch of tucum fibers. Preliminary tensile properties of tucum fibers were obtained from tests conducted in a model 3365 Instron machine (Instron Corp. Norwood, MA, USA) operating at room temperature (~25 °C) with a crosshead speed of 2 mm/min. Each fiber is approximately 1 m long with a non-uniform thickness along its length. Density analysis by geometric method, Archimedes and helium gas pycnometry were performed. To determine the average fiber diameter, 100 random fibers were separated for a statistic analysis, which measured the fiber cross-section dimensions, as well as its length and mass. Five different positions and two different angles (0° and 90°) were investigated in a model Olympus BX53M Microscope (LECO Corporation, St. Joseph, MI, USA). The average linear density and diameter were used to calculate the volume of the tucum fiber $\rho_{g}$:(1)$$\rho_{g}=\frac{M}{\frac{{\pi d}^{2}}{4}\;l}$$
where M is the mass, d the average diameter, and l the length of a fiber specimen.

In the Archimedes test the fiber samples were weighed in air and then submerged in a liquid with lower density than the sample. The weighing process was conducted on an analytical balance with a resolution of 0.0001 g. The apparent density ($\rho_{\mathrm{ap}}$) was obtained by dividing the sample weight in air with sample volume, according to ASTM D3800 standard [30].
(2)$$\rho_{\mathrm{ap}}=\frac{\left(M_{2}-M_{1}\right)\rho_{l}}{((M_{3}-M_{1})-\left(M_{4}-M_{2}\right)}$$
where $\rho_{\mathrm{ap}}$ is the density of liquid (water), M_1_ weight of fiber suspended in air, *M*_2_ the weight of fiber suspended in liquid (to immersion point), M_3_ the weight of suspension the fiber whose density was to be determined in air and M_4_ weight of suspension the fiber whose density was to be determined in liquid. The gas pycnometry was carried out in Ultrapycnometer 1000 by Quantachrome Instruments (Graz, Austria), which aimed to measure the absolute density ($\rho_{\mathrm{aps}}$) of the tucum fiber, following the ASTM D4892 [31] standard. The porosity of the tucum fiber was also investigated through the methods indicated by Luz et al. [32] and Mwaikambo and Ansell [33].

The tucum fiber surface morphology was characterized by scanning electron microscopy (SEM) in a model Quanta FEG 250 Fei microscope (Hillsboro, OR, USA) operating with secondary electrons at 5 kV.

### 2.2. Pullout Tests

Direct methods for estimating interfacial shear strength are based on single-fiber composites. The most direct, the fiber pullout test, involves pulling a partially embedded single fiber out of a block of matrix material [34]. From the resulting tensile stress versus embedded length plot, the interfacial shear strength (τ) and critical length (L_c_) were obtained [35] as follows:(3)$$\tau =\frac{\sigma_{f}d}{2L_{c}}$$
where d is the average equivalent diameter, σ_f_ the tucum fiber tensile strength. Pullout tests were performed in the aforementioned Instron machine with a crosshead speed of 1 mm/min.

### 2.3. Composites Processing

Volume percentages of 20% and 40% tucum fiber in composites were calculated using the rule of mixtures. For the preparation of the composites, the as-received tucum fibers, Figure 2a, were cut and left in an oven at 60 °C for a period of 24 h. The epoxy matrix was prepared by mixing the epoxy diglycidyl ether of bisphenol A (DGEBA) with 13% triethylene tetramine hardener (TETA), both supplied by the company Epoxyfiber (Rio de janeiro, RJ, Brazil).

Composites were prepared by the compression process in a 15 × 12 × 1 cm metallic mold, using different amounts of 20 vol%, Figure 2b, and 40 vol%, Figure 2c, of tucum fiber. The fibers were hand layed in a continuous and aligned manner inside the mold together with the already mixed epoxy resin. As shown in Figure 2b,c, a certain degree of fiber misalignment with respect to the 15 cm dimension (direction) of the model exists. Both composites display around 70% of tucum fibers within less than 5° of misalignment, while a maximum of 35° was calculated in the remaining 30% fibers. This appears to be a common situation for continuous (long) hand layed loose fibers together with fluid polymer resin. A load of 5 ton (3 MPa) was then applied and allowed to cure at room temperature for 24 h.

### 2.4. Tensile Tests

Tensile tests were performed according to the ASTM S3039 standard [36] in the aforementioned Instron machine, with a 25 kN load cell. The test crosshead speed was 2 mm/min. The rectangular 250 × 15 mm samples were cut. For each composition six specimens were tested. The elastic modulus and tensile strength were calculated from the stress-strain curve.

### 2.5. Izod Impact Tests

Izod impact tests were carried out according to the ASTM D256 standard [37]. Composites specimens were cut along the fibers aligned direction. The impact test was conducted on a pendulum model IZ/CH-25 Pantec (Rio de Janeiro, Brazil) operating with a 22 J hammer.

### 2.6. Ballistic Tests

Ballistic tests were performed with .22 ammunition using a compressed air projectile propelled at an average pressure of 4000 psi. Shooting was perpendicular to the target at a distance of 5 m. The apparatus used for the ballistic test is shown in Figure 3. Two ballistic chronographs were placed, one at 10 cm from the end of the specimen and the other at 5 m for determining both projectile speed and impact energy. The absorbed energy (E_abs_) is calculated as a function of the impact (V_i_) and residual (V_r_) velocities of the projectile by:(4)$$E_{\mathrm{abs}}=\frac{1}{2}\;m\;\left(V_{i}^{2}-V_{r}^{2}\right)$$
where m is the .22 projectile mass.

## 3. Results and Discussion

### 3.1. Tucum Fiber Characterization

Table 1 presents preliminary characterization of tucum fiber tensile properties and average diameter as compared to other well-known natural fibers.

As fiber volume fractions and its porosity are important parameters in evaluating the quality of composite materials, both are based on the determination of fiber density. These parameters are necessary in the evaluation of the tucum fiber density [29,30,31,32]. The geometric, Archimedes and gas pycnometry methods were used, and presented an average density of 0.946 g/cm^3^. Measurements from gas pycnometry method, 1.605 g/cm^3^, showed close agreement with literature values as reported by Pennas et al. [29], who found tucum fiber density of 1.51 g/cm^3^. This value is also similar to the density of other NFLs, such as, sisal (1.26–1.50), coir (1.15–1.52), cotton (1.51–1.60) [11].

Figure 4 shows SEM micrographs of the tucum fiber cross-section. It is noteworthy that this cross-section has an irregular elongated shape, which is not constant throughout its length, as illustrated in Figure 4a. Moreover, it has polygonal lumen channels as shown in Figure 4b.

The total, open and closed porosity results, using Luz et al. [32] method, were found to be in the order of 73%, 46% and 27%, respectively. Furthermore, the cellulose content of tucum fiber of 51.8% was calculated using the apparent value, as indicated by Mwaikambo and Ansell [33]. This value of cellulose content is compatible with that found by Pennas et al. [29], as the holocellulose content of the tucum fiber 68.4% comprises not only cellulose but also hemicellulose.

In addition, SEM results contribute to obtain a clear picture of fiber surface morphology. Longitudinal SEM with increasing magnifications is shown in Figure 5. In this figure, one should notice the heterogeneous surface, Figure 5a, with characteristic similar to jute fiber [38] and bamboo fiber [39]. The single fibers are well bonded together to form a larger fiber bundle. Ridges can be clearly seen at regular intervals along the fiber surface, Figure 5b, which may be attributed to fiber overlap within the bundles. Each one of these formations reveals sphere-shaped protrusions, Figure 5c. With higher magnification, Figure 5d, it is noticed that tucum fiber has silica-based spiny protrusions, known as phytoliths by Pennas et al. [29], similar to those found in the piassava fiber [40,41].

In SEM images, Figure 5, a roughness in the surface is observed due to the organic lignin matrix which surrounds the primary cell wall of the fibers, which might contribute to mechanical interlocking and friction forces between the fiber and epoxy matrix. Some natural fibers like piassava [40,41] and coconut [42] also have fine regular array of silicon rich spiny protrusions on their surface. These protrusions are considered as complex surface morphology, which enhances interlocking at the fiber-matrix interface and thereby improving mechanical properties of the composites [41]. Herein, is worth mentioning that the surface condition of a hydrophilic natural fiber plays an important role in its adhesion to a hydrophobic polymer matrix. To improve this surface adhesion several treatments using alkali, silane, acetylation, radiation among others, have been investigated [5,42,43,44,45]. In particular, Battegazzore et al. [46,47] proposed a layer-by-layer nanoengineered technique to modify the surface of natural fibers and produce interphases capable of improving mechanical properties of polymer composites. Recent works used graphene-based materials as a solution to the natural fiber lower adhesion to a polymeric matrix [48]. These new methods of surface modification are now being considered in our ongoing research for application in tucum fiber. However, in the present work, no surface treatment was performed to avoid additional price in the cost effectiveness analysis.

### 3.2. Pullout Tests

Figure 6 shows the results of the pullout tests for different embedded lengths (L) of tucum fiber in the epoxy matrix. A preliminary evaluation of the possibility to perform successful pullout tests led to the observation of two outcomes from the test. Either the fiber breaks before the interface fails or, less frequently, the fiber tip and/or the matrix thin cylinder slips between the grips. As a consequence, a high standard deviation was obtained. In the first stage, for shorter embedded lengths, the tensile strength increases linearly with the embedded length of the fiber in the matrix [49]. As the strength reaches the fiber limit stress, the rupture occurs, which is consistent with the tucum fiber tensile strength in Table 1. The embedded length at which the fiber first fails by pullout is known as critical length (L_c_). For lengths below L_c_ the complete interfacial debonding occurs, while at higher embedded lengths the fiber failure occurs without debonding of the fiber/matrix interface. To calculate the critical length and interfacial strength, the Luz et al. [49] and Monteiro et al. [50] method was used.

The tucum fiber presented a critical length fiber of 6.30 ± 1.33 mm, with interfacial shear strength of 2.73 ± 0.58 MPa, obtained from Equation (3). These values have significant influence on the mechanical properties of fiber reinforced composites. Indeed, for an effective reinforcement the actual fiber length must be longer than the critical fiber length [51]. In the present work, all continuous tucum fibers aligned in the epoxy matrix, Figure 2b,c, have a length of 150 mm, which is much longer than L_c_ and guarantees the highest possible reinforcement [12]. Previous works reported L_c_ for natural fibers such as coir (12.4 mm), PALF (7.3 mm), kenaf (5.37 mm), flax (2.22 mm), jute (0.84 mm) and curaua (0.79 mm) in epoxy matrix. For the interfacial strength the values found for these same natural fibers were: coir (1.42 MPa), PALF (4.93 MPa), kenaf (6.41 MPa), flax (11.83 MPa), jute (11.64 MPa) and curaua (12.93 MPa) [49,50,51,52]. One can observe in Equation (3) that the L_c_ is inversely proportional to the interfacial strength. Hence, the higher the critical length, the weaker the interfacial strength between the fiber and the matrix [53]. In general, a good adhesion between a stiff and strong fiber and the matrix results in a more rigid composite with high tensile strength. In this case, often a brittle impact behavior is observed, because energy absorbing by fiber pullouts are prevented [54]. However, relatively weaker fiber/matrix interactions usually lead to a higher impact resistance due to the better energy absorption by the fiber pullouts [53,54]. A comparison between others natural fibers and synthetic fiber/polymer critical length and interfacial shear strength characterization by pullout test is presented in Figure 7.

### 3.3. Tensile Tests

Table 2. These tensile results revealed that the incorporation, up to 40 vol% of tucum fiber in epoxy matrix increases the ultimate strength by 104%, the elastic modulus by 47%, the total elongation by 36% and the toughness by more than ten times (10×). Furthermore, these results proved that tucum fiber provided higher strength as the tensile stress was absorbed and distributed evenly in the composites. Considering the values shown by Jeyapragash et al. [55], the results of the present work are consistent with those found for other natural fibers reinforcing epoxy matrix.

In comparison to the other natural fibers shown in Figure 8, the tucum fiber epoxy composites present similar performance to others natural fibers epoxy composites of the same origin, such as banana, sisal and areca [55]. However, it displayed inferior performance when compared to epoxy composites with fibers originated from the stem, such as ramie and flax. On the other hand, a higher performance is found in comparison to those of fibers epoxy composites from fruits, such as luffa, groundnuts and coir [55].

SEM images after tensile test of tucum fiber reinforced epoxy composites are shown in Figure 9. In Figure 9a one can see the fiber/matrix interface adhesion after tensile test and also the matrix river marks, this being a typical characteristic of fragile polymer fracture. Fiber pullout and broken fibers are observed in Figure 9b due to fiber extraction and debonding. The fiber failure mode gives an indication of reasonable load transfer through fibers and matrix. In contrast, a weak fiber/matrix interaction, in which load transfer from the matrix to the fiber is not effective, leads to a reduction in the tensile properties, and may also result in a decrease in the impact strength due to the absence of friction during the fiber pullouts [51,56,57]. Depending on the reinforcing fiber, the strength of a composite can also be reduced by weak fiber/matrix adhesion.

### 3.4. Izod Impact Tests

Figure 10 presents the variation of absorbed energy and fracture aspects of the composites, with 2 and 40 vol% of tucum fiber, fractured after Izod impact test.

In the Izod impact tests it was verified that both 20 vol% and 40 vol% of tucum fiber acted as reinforcement in the epoxy matrix. This can be seen by the macrographs, Figure 10a,b, that indicates the participation of the fibers in the composite fracture. In addition, the graph in Figure 10c shows that there is a 157% increase in the average energy absorption of the 40 vol% (216 J/m) composite in comparison with neat epoxy (0%) (84 J/m). This behavior was expected for NLFs since the higher the volume fraction of fibers present in the composite, the greater the impact energy needed to break the Izod specimens. Based on the average energy absorption value in Figure 10c, an exponential adjustment of the data obtained for tucum fiber composites was performed. The exponential curve that represents the impact resistance behavior of these composites as a function of the volume fraction of tucum fiber is also shown in Figure 10c with the respective mathematical equation. These Izod impact tests results are consistent with the tensile toughness results in Table 2.

Figure 9 shows a comparison between the data of the present work and the results of epoxy matrix composites reinforced with different NLFs, as well as of plain epoxy obtained in previous works [20,58,59,60,61]. Comparing the results of epoxy matrix composites reinforced with different NLFs with the composites investigated in the present work, it was found that epoxy composites with 20 vol% of mallow fiber and 20 vol% of ramie fibers exhibit 3 and 2 times higher, respectively, average absorption energy than the tucum fiber composites. While the composite with 40 vol% of tucum fiber absorbed an average impact energy similar to the composite with 40 vol% of fique fabric.

Table 2. 73 MPa, which leads to lower toughness. As such, it decreases the area of fracture and, consequently, a lower energy is absorbed in the hammer impact. Margem et al. [62] found for mallow fiber/epoxy the critical length of 2.6 mm, and an interfacial strength of 3.1 MPa, these results are comparable to the present case, whereas Costa et al. [60] found a higher value for Izod impact energy in association with lower mallow fiber/epoxy interfacial strength, presented in Figure 11. These previous results evidence that the main mechanism of fracture in Izod impact test is the pullout, which was reported to be the main contributor to the total absorbed impact energy. It should also be mentioned that polyester composites were reported to display comparable Izod impact absorbed energy and fracture mechanism with that for epoxy composite in the case of hemp fiber reinforcement [63].

### 3.5. Ballistic Tests

Table 3 shows the results of the ballistic tests. In these tests, five shootings were made for each composite and it was observed that for both composites with 20 vol% and 40 vol% of tucum fiber occurred total perforation of the plates. Based on the results of Table 1, there is a reduction in the residual velocity of the projectile for the composite with 20 vol% of tucum fiber. Hence, this composite exhibited greater capacity than the 40 vol% for energy absorption. A possible justification for this result is the predominant performance of the brittle behavior of the epoxy matrix. Indeed, the plain epoxy shows the highest absorbed energy. However, due to its brittle behavior, as a ballistic armor, a neat epoxy is completely fragmented and would not be used for personal (body) protection against subsequent shootings, as required by the standard [64]. By contrast, both 20 vol% and 40 vol% tucum fiber epoxy composites suffered only perforation but not total fragmentation, as illustrated in Figure 3b.

Although showing not as expressive greater values for the tested properties, tucum fiber epoxy composite can be competitive in terms of specific properties, as well as cost. This topic is one of the important reasons for the increasing use of improved cost effectiveness natural fiber, instead of synthetic fiber. In this aspect, Shahinur and Hasan [65] reported that natural fibers have better performance and cost effectiveness compared to synthetic fibers as well as to their composites. The tucum fiber can be found in the local market, in bundle format, ranging in weight from 100 to 300 g/bundle, with a price of approximately USD 8.68/kg, similar to the jute fabric price (USD 8.38) [66]. This price is higher than that found for mallow (R$ 2.65/kg—08/2018) [67], sisal (R$ 2.59/kg) [67], flax (USD 0.67/kg) [68], fique fibers (USD 0.45/kg) [61,69]. While the price of the composite is similar to the values found for composites of natural fibers, it is much inferior to that of the composites with synthetic aramid and glass fibers, presented in Table 4.

## 4. Conclusions

Tucum fibers extracted from the leaf-stalk of the Amazon *Astrocaryum vulgare* palm tree were evaluated as possible reinforcement of novel epoxy composites.Measurements from geometry, Archimedes and gas pycnometry methods of the tucum fiber showed an average value of 0.946 g/cm^3^. Moreover, the total, open and closed porosity results were 73%, 46% and 27%, respectively.Pullout tests revealed that the critical length of the tucum fiber in the epoxy matrix is 6.30 mm, and the interfacial shear stress 2.73 MPa.Tensile tests showed for the first time that 20 vol% and 40 vol% of tucum fibers act as effective reinforcement for the epoxy matrix increasing 104% the strength, 47% the stiffness and more than 10 times the toughness of the composite.As compared to the plain epoxy, the tucum fiber composite displays an exponential increase in Izod impact energy with fiber volume fraction. The composite with 40 vol% of tucum fiber incorporated in the epoxy matrix exhibited an 157% increase in the average absorbed energy.The Izod impact energy absorbed by the samples reinforced with 20 vol% tucum was less than those reinforced with fiber of mallow and ramie, tested in previous works, while the composite with 40 vol% tucum had a performance similar to the composite with 40 vol% fique fabric.In the ballistic test, the composite with 20 vol% of tucum fiber exhibited greater energy absorption capacity, 20%, in comparison to 40 vol% tucum fiber composite, which can be justified by the predominance of the fragile behavior of the epoxy matrix. In contrast to neat epoxy, these composites are not fragmented after ballistic test.The cost of the tucum fiber epoxy composites is the lowest among natural fibers composites and much inferior to those of synthetic fiber composites. As such, these novel composites might be applied as substitute for conventional composites reinforced with either natural or synthetic fibers, particularly in multilayered armor for body protection.

## Figures and Tables

**Figure 1 polymers-12-02259-f001:**
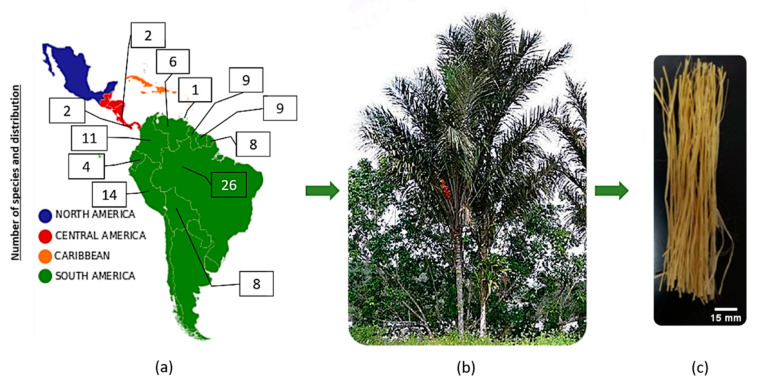
The palm genus *Astrocaryum* endemic in (**a**) South America and Central America, (**b**) Tucum palm tree and (**c**) fiber (*Astrocaryum vulgare* chosen for present work).

**Figure 2 polymers-12-02259-f002:**
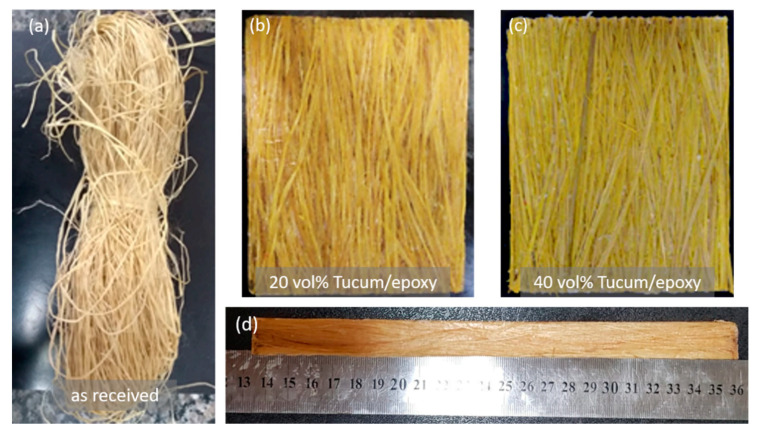
(**a**) Tucum fiber as-received, and epoxy composite plates with continuous and aligned tucum fiber in different volume fractions: (**b**) 20%; (**c**) 40%; (**d**) upper view.

**Figure 3 polymers-12-02259-f003:**
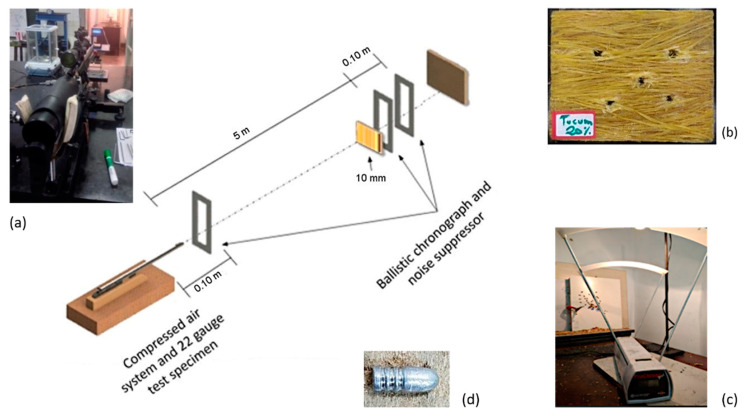
Schematic of apparatus used in the ballistic test: (**a**) compressed air gun; (**b**) perforated composite sample; (**c**) chronographs; and (**d**) .22 ammunition.

**Figure 4 polymers-12-02259-f004:**
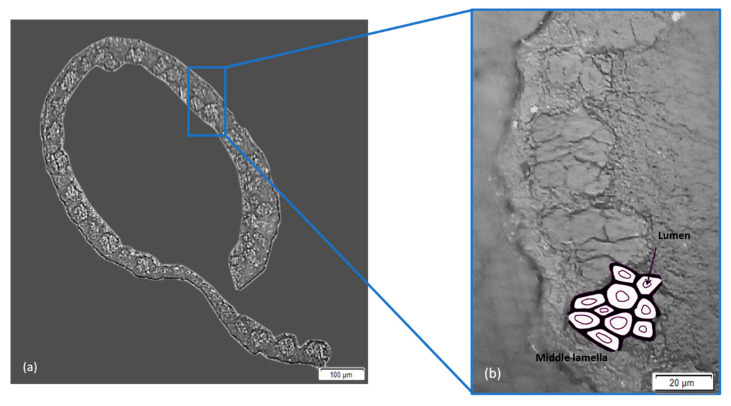
Cross-section morphology of the tucum fiber bundle (**a**) 10× and (**b**) 50×.

**Figure 5 polymers-12-02259-f005:**
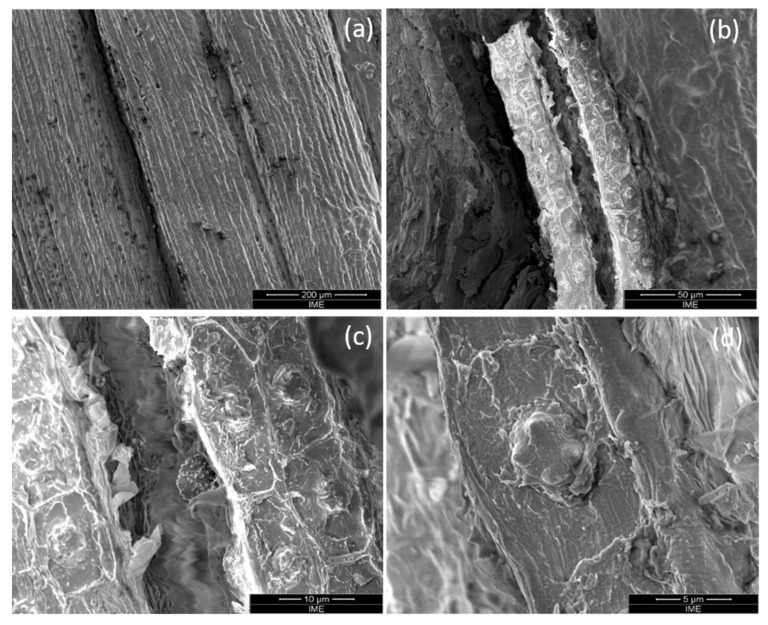
Surface morphology of tucum fiber with increasing magnification (**a**) 400×; (**b**) 1600×; (**c**) 6000×; (**d**) 12000×.

**Figure 6 polymers-12-02259-f006:**
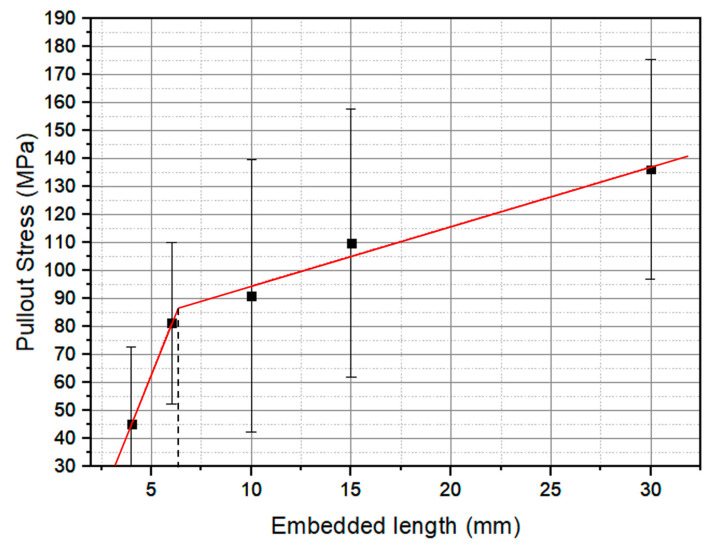
Apparent interfacial pullout strength (MPa) between Epoxy and tucum fibers as a function of embedded length (mm).

**Figure 7 polymers-12-02259-f007:**
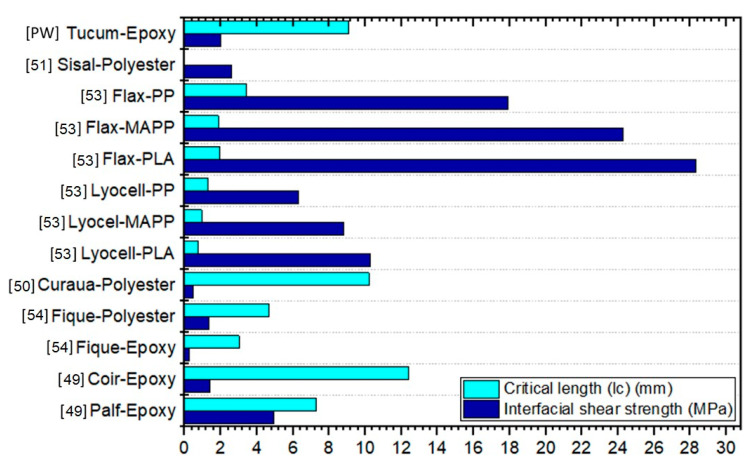
Comparison of pullout tests of different natural fibers/polymer. Palf-epoxy and coir-epoxy [49]; lyocell-PLA, lyocel-MAPP, lyocell-PP, flax-PLA, flax-MAPP, flax-PP [53]; sisal-polyester [51]; curaua-polyester [50], fique-epoxy and fique-polyester [54]; tucum-epoxy (Present work (PW)).

**Figure 8 polymers-12-02259-f008:**
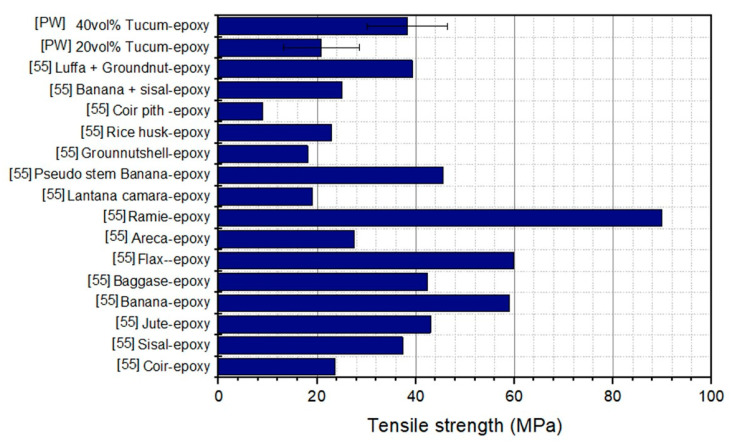
Tensile strength comparison between tucum fiber (Present work (PW)) and others natural fiber epoxy composites.

**Figure 9 polymers-12-02259-f009:**
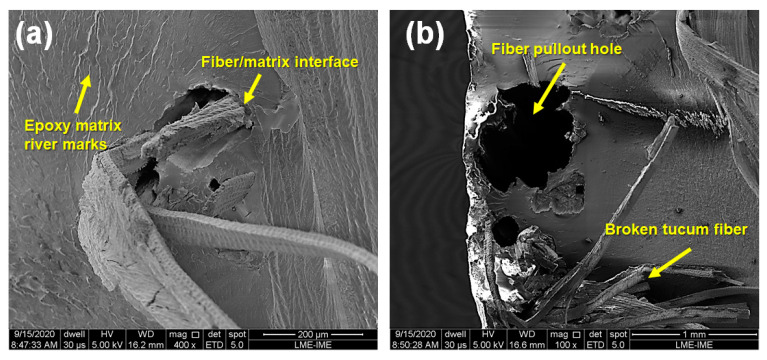
Composite with 40 vol% of tucum fiber with different magnifications: (**a**) 400× and (**b**) 100×.

**Figure 10 polymers-12-02259-f010:**
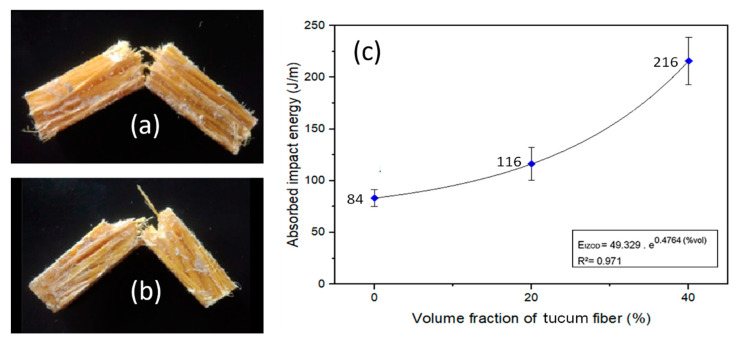
Composites fractured by impact Izod: (**a**) 20 vol%, and (**b**) 40 vol% of tucum fiber/epoxy, and (**c**) variation of the absorbed impact energy with volume fraction of tucum fiber.

**Figure 11 polymers-12-02259-f011:**
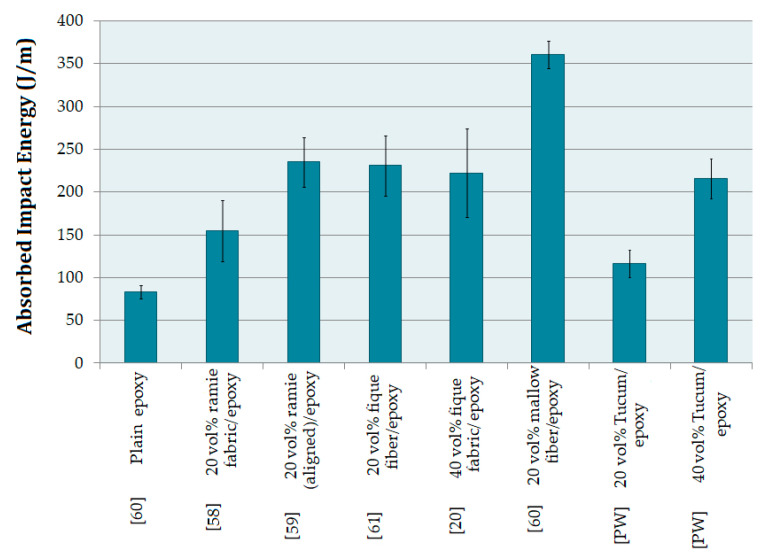
Energy absorbed by epoxy-NLFs composites by the Izod impact for several epoxy composites reinforced by natural fibers and fabrics (Present Work (PW)).

**Table 1 polymers-12-02259-t001:** Preliminary tensile properties and diameter of tucum as compared to others well-known natural fibers.

Natural Lignocellulosic Fiber	Tensile Strength (MPa)	Elastic Modulus (GPa)	Total Elongation (%)	Diameter (mm)	Reference
Tucum	124	5	2.10	0.51	PW
Sisal	287	9	2.50	0.14	[11]
Coir	95	4	1.60	0.25	[11]
Jute	393	13	3.00	0.07	[11]
Hemp	389	35	3.40	0,03	[11]

PW = Present Work.

**Table 2 polymers-12-02259-t002:** Results for tensile test of the tucum fiber epoxy composites.

Fiber Volume (%)	Tensile Strength (MPa)	Elastic Modulus (GPa)	Total Elongation (%)	Tensile Toughness (J/mm^3^)
0	18.7 ± 7.6	1.7 ± 0.1	1.1 ± 0.4	6.6 ± 4.3
20	20.8 ± 7.7	1.9 ± 0.4	1.0 ± 0.2	14.3 ± 6.2
40	38.3 ± 8.1	2.5 ± 0.4	1.5 ± 0.4	75.5 ± 43.2

**Table 3 polymers-12-02259-t003:** Ballistic test data with .22 ammunition.

Composite	Projectile Mass (g)	Initial Velocity (m/s)	Residual Velocity (m/s)	Average Absorbed Energy (J)
Neat epoxy	3.35 ± 0.05	290.55 ± 11.22	140.82 ± 5.36	108.20 ± 12.38
20 vol% fiber	3.27 ± 0.05	266.34 ± 23.14	139.78 ± 8.85	84.02 ± 8.42
40 vol% fiber	3.29 ± 0.07	285.38 ± 16.87	197.82 ± 13.09	69.60 ± 9.14

**Table 4 polymers-12-02259-t004:** Cost comparison of composites with natural and synthetic fibers.

Composite Material	Cost (US Dollars)	Reference
30 vol% jute fabric/epoxy	4.06	[66]
15 vol% fique fabric/epoxy	4.87	[70]
40 vol% fique fabric/epoxy	3.67	[70]
64.8 vol% aramid laminate/epoxy	49.59	[71]
72 vol% glass fiber/epoxy	18.06	[71]
20 vol% tucum fiber/epoxy	1.78	PW ^1^
40 vol% tucum fiber/epoxy	1.74	PW ^1^

^1^ PW: Present Work.

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
