# Peer review of "Tucum Fiber from Amazon Astrocaryum vulgare Palm Tree: Novel Reinforcement for Polymer Composites"

_polymers, 2020, doi:10.3390/polym12102259_

Round 1

Reviewer 1 Report

This manuscript used Tucum fiber and epoxy resin to fabricate polymer composite, and carried out fiber pullout test, Izod impact test, and ballistic test. I think the experimental results are not sufficient, and major revision should be made.

Q1. The tensile properties of Tucum fiber should be measured and compared with other plant fibers.

Q2. The authors should add the cross-sectional morphology of Tucum fiber composite.

Q3. In table 1 the initial velocity of neat epoxy is much higher than 20 vol% tucum/epoxy composite, while the residual velocities are close. Why is the average absorbed energy of neat epoxy lower than 20 vol% tucum/epoxy? How did the authors calculate the average absorbed energy?

Q4. In Figure 2, Tucum fiber seems not to totally align. How much is the degree of alignment?

Q5. Based on the results in figure 6, I think the critical length should be a range.

Q6. Is there any potential method to treat the surface of Tucum fiber?

Q7. For ballistic test, cross-ply laminate may be further measured.

Author Response

The authors would like to thank the Reviewer for the valuable comments and suggestions that contribute to improve our manuscript. Amendments were provided accordingly and all modifications were marked as Track Changes in the revised version of the manuscript. Responses to each comment, point by point, are given below.

General Comment: This manuscript used Tucum fiber and epoxy resin to fabricate polymer composite, and carried out fiber pullout test, Izod impact test, and ballistic test. I think the experimental results are not sufficient, and major revision should be made.;

Response: The authors agree with the reviewer, and new recommended experimental results, such as tensile properties and SEM analysis, as well as major revision are provided in the revised version.

Q1. The tensile properties of Tucum fiber should be measured and compared with other plant fibers.;

Response: Tensile properties of Tucum fiber are now included and compared with others fibers.

Q2. The authors should add the cross-sectional morphology of Tucum fiber composite.;

Response: As requested, Tucum fiber composite cross-sectional morphology is now added and discussed.

Q3. In table 1 the initial velocity of neat epoxy is much higher than 20 vol% tucum/epoxy composite, while the residual velocities are close. Why is the average absorbed energy of neat epoxy lower than 20 vol% tucum/epoxy? How did the authors calculate the average absorbed energy?;

Response: These are relevant questions, with explanation given in terms of the specific ballistic mechanism of bullet impact energy absorption by the Tucum composite. The calculation of average absorbed energy is now presented.

Q4. In Figure 2, Tucum fiber seems not to totally align. How much is the degree of alignment?;

Response: Indeed, the Tucum fibers are not totally aligned inside the epoxy matrix due to the hand layup process. The degree of alignment is now calculated and indicated in the revised version.

Q5. Based on the results in figure 6, I think the critical length should be a range.;

Response: The reviewer is correct, the critical length and, consequently, the interfacial strength are within a range, which is now indicated.

Q6. Is there any potential method to treat the surface of Tucum fiber?;

Response: Excellent question, our ongoing work on ballistic performance of Tucum fiber composites against heavy rifle ammunition (caliber 7.62 mm) is considering fiber surface treatment (functionalization) with graphene oxide.

Q7. For ballistic test, cross-ply laminate may be further measured.;

Response: In fact, we are already preparing cross-ply laminate samples in our ongoing research to improve high impact ballistic performance and avoid delamination associated with a standard required sequence of shootings.

Reviewer 2 Report

This manuscript deals with the use of tucum fiber for the production of biocomposites employing epoxy as a matrix. I found this manuscript potentially interesting and worth consindering for publication after the following modifications:

  1. Introduction, as mentioned by the authors the use of natural fibers for the production of composites is increasing and there is an increasing trend in reported research trying to improve the interfacial strength of these composites, it would be thus interesting to cite recent literature (Es Composites Part B: Engineering, 2020, 108310, ACS Sustainable Chemistry & Engineering, 2018 6 (8), 9601-9605 )
  2. the reported images show low quality, this is particularly important for micrographs were by zooming in a lot of details are lost. In addition it would be interesting to see low magnification images of the fibers.
  3. Figure 6 shows a rather big standard deviation for the measured values. Is this related to the natural nature of the fibers ? can the authors further discuss this ?
  4. Figure 8 needs error bars added. In this section, the authors should explain the effects of natural fibers on the increased impact resistance, is this related to the extra work associated to fiber pull-out ? this should be documented by imaging the fracture of the composites after the tes
  5. table 1 needs the error added. In addition the authors should give more details about this ballistic test describing its relevance to the material properties and whether this could be related to a possible application of the composites.

Author Response

The authors would like to thank the Reviewer for the valuable comments and suggestions that contribute to improve our manuscript. Amendments were provided accordingly and all modifications were marked as Track Changes in the revised version of the manuscript. Responses to each comment, point by point, are given below.

General Comments: This manuscript deals with the use of tucum fiber for the production of biocomposites employing epoxy as a matrix. I found this manuscript potentially interesting and worth considering for publication after the following modifications.;

Response: The authors appreciate the kind words regarding our manuscript and have attended all recommended modifications. 

Comment 1: Introduction, as mentioned by the authors the use of natural fibers for the production of composites is increasing and there is an increasing trend in reported research trying to improve the interfacial strength of these composites, it would be thus interesting to cite recent literature (Es Composites Part B: Engineering, 2020, 108310, ACS Sustainable Chemistry & Engineering, 2018 6 (8), 9601-9605).;

Response: Indeed, we find quite relevant to cite the indicated recent literature.

Comment 2: The reported images show low quality, this is particularly an important for macrographs were zooming in a lot of details are lost. In addiction it would be interesting to see low magnification images of the fibers;

 Response: Better quality images are now shown in the revised version with added low magnification images of the Tucum fiber.

Comment 3: Figure 6 shows a rather big standard deviation for the measured values. Is this related to the natural nature of the fibers? can the authors further discuss this ?;

Response: These are important questions that are now explained in terms of the variability of natural fibers and discussed accordingly.

Comment 4: Figure 8 needs error bars added. In this section, the authors should explain the effects of natural fibers on the increased impact resistance, is this related to the extra work associated to fiber pull-out ? this should be documented by imaging the fracture of the composites after the tes;

Response: Error bars are now introduced in Figure 8. The effect of Tucum fiber on the increased impact resistance of epoxy composites is now explained based on the interfacial shear strength obtained in the pullout tests. Composites fracture images are documenting this effect.

Comment 5: Table 1 needs the error added. In addition the authors should give more details about this ballistic test describing its relevance to the material properties and whether this could be related to a possible application of the composites.;

Response: Errors are added in values presented in new Table 3 (old Table 1). As requested, more details on the ballistic test, as well as its relevance to the composite’s properties, including possible applications, are presented in the new revised version.

Round 2

Reviewer 2 Report

the authors have answered my previous queries. just a minor comment:

please make sure that the quality of the uploaded images is the highest possible. the figures of the manuscript appear to be  low definition images in the available pdf